# Shedding Light on the Role of Na,K-ATPase as a Phosphatase during Matrix-Vesicle-Mediated Mineralization [note 1]

**DOI:** 10.3390/ijms232315072

**Published:** 2022-12-01

**Authors:** Heitor Gobbi Sebinelli, Luiz Henrique Silva Andrilli, Bruno Zoccaratto Favarin, Marcos Aantonio Eufrasio Cruz, Maytê Bolean, Michele Fiore, Carolina Chieffo, David Magne, Andrea Magrini, Ana Paula Ramos, José Luis Millán, Saida Mebarek, Rene Buchet, Massimo Bottini, Pietro Ciancaglini

**Affiliations:** 1Departamento de Química, Faculdade de Filosofia, Ciências e Letras de Ribeirão Preto da Universidade de São Paulo (FFCLRP-USP), Ribeirão Preto, São Paulo 14040-900, Brazil; 2University Lyon, Université. Claude Bernard Lyon 1, CNRS UMR 5246, ICBMS, F-69622 Lyon, France; 3Department of Biomedicine and Prevention, University of Rome Tor Vergata, 00133 Rome, Italy; 4Sanford Burnham Prebys, La Jolla, CA 92037, USA; 5Department of Experimental Medicine, University of Rome Tor Vergata, 00133 Rome, Italy

**Keywords:** Na,K-ATPase, liposome, matrix vesicle, mimetic model, biomineralization, apatite

## Abstract

Matrix vesicles (MVs) contain the whole machinery necessary to initiate apatite formation in their lumen. We suspected that, in addition to tissue-nonspecific alkaline phosphatase (TNAP), Na,K,-ATPase (NKA) could be involved in supplying phopshate (P_i_) in the early stages of MV-mediated mineralization. MVs were extracted from the growth plate cartilage of chicken embryos. Their average mean diameters were determined by Dynamic Light Scattering (DLS) (212 ± 19 nm) and by Atomic Force Microcopy (AFM) (180 ± 85 nm). The MVs had a specific activity for TNAP of 9.2 ± 4.6 U·mg^−1^ confirming that the MVs were mineralization competent. The ability to hydrolyze ATP was assayed by a colorimetric method and by ^31^P NMR with and without Levamisole and SBI-425 (two TNAP inhibitors), ouabain (an NKA inhibitor), and ARL-67156 (an NTPDase1, NTPDase3 and Ecto-nucleotide pyrophosphatase/phosphodiesterase 1 (NPP1) competitive inhibitor). The mineralization profile served to monitor the formation of precipitated calcium phosphate complexes, while IR spectroscopy allowed the identification of apatite. Proteoliposomes containing NKA with either dipalmitoylphosphatidylcholine (DPPC) or a mixture of 1:1 of DPPC and dipalmitoylphosphatidylethanolamine (DPPE) served to verify if the proteoliposomes were able to initiate mineral formation. Around 69–72% of the total ATP hydrolysis by MVs was inhibited by 5 mM Levamisole, which indicated that TNAP was the main enzyme hydrolyzing ATP. The addition of 0.1 mM of ARL-67156 inhibited 8–13.7% of the total ATP hydrolysis in MVs, suggesting that NTPDase1, NTPDase3, and/or NPP1 could also participate in ATP hydrolysis. Ouabain (3 mM) inhibited 3–8% of the total ATP hydrolysis by MVs, suggesting that NKA contributed only a small percentage of the total ATP hydrolysis. MVs induced mineralization via ATP hydrolysis that was significantly inhibited by Levamisole and also by cleaving TNAP from MVs, confirming that TNAP is the main enzyme hydrolyzing this substrate, while the addition of either ARL-6715 or ouabain had a lesser effect on mineralization. DPPC:DPPE (1:1)-NKA liposome in the presence of a nucleator (PS-CPLX) was more efficient in mineralizing compared with a DPPC-NKA liposome due to a better orientation of the NKA active site. Both types of proteoliposomes were able to induce apatite formation, as evidenced by the presence of the 1040 cm^−1^ band. Taken together, the findings indicated that the hydrolysis of ATP was dominated by TNAP and other phosphatases present in MVs, while only 3–8% of the total hydrolysis of ATP could be attributed to NKA. It was hypothesized that the loss of Na/K asymmetry in MVs could be caused by a complete depletion of ATP inside MVs, impairing the maintenance of symmetry by NKA. Our study carried out on NKA-liposomes confirmed that NKA could contribute to mineral formation inside MVs, which might complement the known action of PHOSPHO1 in the MV lumen.

## 1. Introduction

Biomineralization consists of the accumulation of minerals, composed mainly of phosphate and calcium ions that form hydroxyapatite (Ca_10_(PO_4_)_6_(OH)_2_) or related carbonated apatites when the hydroxyls are substituted by other anions and propagate them onto the collagenous matrix. The ossification process mediated by osteoblasts (in the formation of flat bones) or by odontoblasts (in tooth formation) is clearly distinct from that which occurs during the calcification of epiphyseal cartilage (endochondral formation) which is mediated by hypertrophic chondrocytes [1]. In all of these tissues, mineralization depends on the homeostasis of calcium (Ca^2+^) and inorganic phosphate (P_i_) ions, the adequate protein composition of the extracellular matrix, and the absence and/or removal of mineralization inhibitors [1]. The initiation of mineral formation by matrix vesicles (MVs) in the growth plate cartilage is supported by several pieces of evidence [2,3,4,5,6,7,8,9,10,11]. (i) In the early 1960s, lipids were discovered at the sites of calcification in the growth plate [12,13,14,15]. Later, it was evidenced by transmission electron microscopy that the lipids formed vesicular structures, with diameters in the 100–300 nm range [16,17,18,19], named MVs [2] that were able to bind strongly to collagen [20,21]. (ii) MVs were shown to be released from the polarized apical side of plasma membranes of hypertrophied chondrocytes [22]. (iii) and to possess the ability to accumulate calcium and phosphate ions in their lumen [23,24,25,26,27,28,29], which can nucleate [30,31,32,33] and form apatites [29]. (iv) Once released from MVs, either by mechanical force and/or degradation of phospholipids [22,34,35,36,37,38], these apatitic crystals continue to grow onto collagen fibers in the extracellular matrix [2,3]. MVs released from osteoblasts and odontoblasts have similar properties and are also able to initiate apatite within their lumen [2,3,10]. 

MVs harbor the complete machinery necessary to sustain the supply of P_i_ in the lumen of MVs and to control the homeostasis of extracellular inorganic pyrophosphate (PP_i_) [2,3,4,5,6,7] (Figure 1). PP_i_ inhibits the formation of apatite crystals [39] as evidenced by hypomorphic mutations of tissue-nonspecific alkaline phosphatase (TNAP) gene (*ALPL*), which impair PP_i_ hydrolysis and lead to the accumulation of extracellular PP_i_ that cause the soft bones disease known as hypophosphatasia [40,41]. The function of MVs is to accumulate P_i_ and Ca^2+^ inside their lumen, which provides an optimum environment to nucleate and form apatite and to release apatite in the extracellular medium, where the PP_i_ has to be removed to sustain the second step of mineral formation [2,3,4,5,6,7,10] (Figure 1). High levels of TNAP, the enzyme responsible for PP_i_ hydrolysis, have been found in osteoblast membranes during bone tissue mineralization [2,42,43,44] and in MVs [28,45,46]. The enzyme is allosterically regulated by ATP [47] and competitively inhibited by the reaction product, P_i_ [48], suggesting that the relative levels of PP_i_, an inhibitor of biomineralization, and P_i_, a promoter of mineralization but also an inhibitor of TNAP, present in the matrix extracellular fluid also play an important role in regulating the physiological mineralization process. 

Another extracellular phosphatase is Nucleotide Pyrophosphataes/Phosphodiesterase 1(NPP1), which produces PP_i_ and P_i_ from ATP [39] (Figure 1). NPP1, also known as glycoprotein-1 (PC-1), is bound to the plasma membrane, while another member of the NPP family, autotaxin (NPP2), is secreted, and B10 (NPP3) is abundant in intracellular spaces [39]. These three isoenzymes are expressed in a wide variety of tissues, including bone and cartilage [40], and all have the common ability to hydrolyze diesters of phosphoric acid to phosphomonoesters, primarily ATP to AMP and/or ADP to adenosine [41]. NPP1, which has its active site facing the extracellular milieu, is found in high concentrations on the surfaces of osteoblasts and chondrocytes, as well as in the membranes of their MVs [42,43] (Figure 1). NPP1 inhibits apatite precipitation by its PP_i_-generating property. This hypothesis has been confirmed by in vitro studies, where cells transfected with the NPP1 cDNA demonstrated high levels of PP_i_ in osteoblast-derived MVs, accompanied by reduced matrix mineralization [43,44]. Abnormal precipitation of Ca^2+^ pyrophosphate dehydrate (CPPD) has been observed in association with TNAP deficiency. While excess PP_i_ in patients with hypophosphatasia primarily leads to rickets and osteomalacia, patients with the disease can also develop the pathological formation of CPPD crystals in articular cartilage [45]. NPP1 also acts as a phosphatase in the absence of TNAP [41,46,47,48,49]. The nucleoside triphosphate diphosphohydrolase 1 (CD39) also hydrolyses extracellular ATP or ADP forming AMP with the release of one P_i_ ion [50]. 

On the other hand, P_i_ needs to be concentrated or generated to sustain the formation of apatitic minerals inside MVs. P_i_ is thought to accumulate into MV via a Na^+^-dependent P_i_ transport system [26], although the substitution of Li^+^ or K^+^ for Na^+^ has a minimal effect [27]. One likely P_i_ transporter could be the phosphate transporter 1 (P_i_T-1) [28] (Figure 1). Alternatively, there are enzymes that hydrolyze phosphoesters, including phosphocholine and phosphoethanolamine, inside MVs to generate P_i_. High levels of PHOSPHO1 in MVs isolated from chondrocytes were revealed by immunoblotting [51] (Figure 1). By comparing wild-type and *Phospho1*^−/−^ mice, it has been shown that the function of PHOSPHO1, another prominent phosphatase present in MVs, is to produce intraluminal P_i_ and induce apatite mineral [52]. Nevertheless, the gradient of Na^+^ generated by Na,K-ATPase (NKA), along with its own ATP hydrolysis, could be theoretically sufficient to produce a burst of P_i_ inside of the MVs to form the nucleational core, using the auxiliary action of P_i_T-1 cotransporter [53,54] (Figure 1).

NKA belongs to the P-Type-II ATPase family; it is ubiquitous at the cellular level for animal tissues and vital for these organisms [55]. The functions performed by the protein in the cells are related to the maintenance of the ionic composition of the cytosolic medium [56]. Thus, several tissues use NKA to carry out the transport of three Na^+^ ions out of the cell and two K^+^ ions into the cell through the plasma membrane with a perpetual energy expenditure of one ATP molecule [57]. This uneven movement of ionic transport generates an electrolytic gradient that allows the cell to reuse energy by propelling secondary transport of molecules such as sugars, neurotransmitters, metabolite amino acids, and other H^+^, Ca^2+^, Cl^−^, and other ions [57]. 

To assess NKA contribution in the production of luminal P_i_ in MVs, ATP hydrolysis of MVs extracted from epiphyseal cartilage from femurs of chicken embryos was determined on solubilized MVs by using Levamisole (a TNAP inhibitor of low specificity), and SBI-425 (a TNAP inhibitor of high specificity) [58,59] ouabain (an NKA inhibitor) and without any inhibitors. The ability to induce apatite formation was assessed in native MVs and in MVs devoid of TNAP through cleavage of their GPI anchor moiety. Additionally, NKA reconstituted in liposomes was used as a biomimetic model to study the role of NKA during mineral formation. In this respect, the advantage of a proteoliposome system is to have control over lipid and protein composition and observe NKA activity in isolation [60,61]. NKA was solubilized by using a non-ionic detergent and reconstituted in liposomes containing a 1:1 (*w*/*w*) mixture of DPPC:DPPE in a 1:3 (*w*/*w*) lipid:protein ratio resulting in proteoliposomes with 89% recovery of incorporated protein, with a mean diameter of 140 nm and 79% recovered ATPase activity. The reconstituted NKA has the ATP hydrolysis site located on the outside of the proteoliposome vesicles, called inside-out orientation [62,63]. The inside-out proteoliposomes allow a series of biophysical studies on enzyme stability and structure. Although the functional unit of NKA in vivo is not yet clearly elucidated, it is known that the oligomerization process occurs in natural membranes, and it cannot be ruled out that this behavior may be a natural mechanism developed by organisms to act in the metabolic regulation of various functions in the plasma membrane of cells [64,65]. Atomic force microscopy (AFM) was applied in intermittent contact mode to obtain morphological and topographical information on how NKA is organized in proteoliposomes composed of DPPC and DPPC:DPPE (1:1 molar ratio), to assess the accessibility of the active site and to optimize mimetic models to match their behavior of biological systems [66].

## 2. Results and Discussion

### 2.1. Characterization of MV

Twenty-five different fresh MV samples were extracted, as reported by [66]. The average protein concentration of MVs was around 2.9 ± 0.9 mg·mL^−1^. The specific TNAP activity of MVs was 9.2 ± 4.6 U·mg^−1^, which was significantly higher than that of plasma membranes of chondrocytes which amounted to 5.1 ± 3.6 U·mg^−1^, consistent with the significant reported enrichment of TNAP in MVs compared to plasma membranes [66]. The mean diameter of 10 distinct samples of MVs was 212 ± 19 nm, as revealed by DLS, which falls within the reported 100–300 nm diameter range [16,17,18,19]. However, DLS analysis revealed a higher polydispersity index (PI) than the ideal for a monodisperse solution, suggesting the presence of a mixture of different-sized vesicles (Figure 1). A mean diameter of 180 ± 85 nm was obtained (Figure 2), in accordance with that revealed by Dynamic Light Scattering (DLS) analysis and with AFM values found in the literature [29]. This technique allows the investigation of surface morphology from nanostructures or microdomains composed of lipids and proteins. The phase contrast shift became more evident when using synthetic vesicles such as liposomes and proteoliposomes but was not evident in the MV surface [29,66]. The sum of the infinite interferences arising from the unlimited variety of proteins and lipids probably dilutes the equipment signal and reduces the visualization of structures on the surface. Thus, in the phase images, the MVs have low-range phase shifts (Figure 2B, color scale). Even protrusions, a characteristic of the several proteins present in the membrane, are difficult to be observed (Figure 2A, 3D topographic image).

### 2.2. ATP Hydrolysis by MV 

In order to comprehend the role of NKA in MVs, ATP hydrolysis was measured by ^31^P NMR and a colorimetric assay. To determine the ATPase activity inside and outside MVs, the MVs were solubilized by the addition of 0.2% of C_12_E_8_ or Nonidet P-40. MVs without detergent have a mean diameter of 206 nm and PI of 0.14, while MVs in the presence of the detergent resulted in a diameter of 79.9 nm and PI of 1.82. Therefore, MVs were destabilized by the addition of the detergent, exposing ATP to all regions of MVs and enabling the measurement of ATP hydrolysis inside MVs. The addition of 2 mM of ouabain was sufficient to inhibit 90% of the total activity of NKA and 99.9% of the activity when used at 5.0 mM [62,63]. The addition of 3 mM of ouabain in MVs resulted in a very small decrease in total ATP hydrolysis to around 3–7% (Table 1), which was consistent with the small ATP hydrolysis by ATPase in MVs using [γ-^32^P]-ATP and P_i_/molibdate complex determination [53,67]. 

Levamisole and SBI-425 were more efficient in inhibiting ATP hydrolysis as they act mainly and exclusively, respectively, on TNAP activity (Table 1). The addition of 5 mM of Levamisole, the stereoisomer of tetramisole, a potent uncompetitive inhibitor [68], led to a decrease of around 72–69% of the total ATP hydrolysis by MVs (Table 2) compared to the control experiment. SBI-425 is a potent pharmacological uncompetitive TNAP inhibitor that, unlike Levamisole, acts specifically on TNAP without cross-inhibition of other phosphatases [58,59]. Two concentrations of SBI-425 (5 μM and 10 μM), at much lower concentrations than those used for Levamisole, resulted in inhibitions of 21.5 and 35.4%, respectively (Table 1), which confirmed that TNAP is hydrolyzing ATP. 

6-N,N-Diethyl-D-beta-gamma-dibromomethylene adenosine triphosphate (ARL-67156) is a competitive inhibitor of the ectoenzymes NTPDase1 and NTPDase3 along with NPP1 [70]. The inhibition of CD39 (an NTPDase1) activity has been reported to induce the osteogenic potential of mesenchymal stem cells derived from gingival tissue by regulating the balance between osteoclasts and osteoblasts through the Wnt/β-Catenin pathway in an osteopenic model induced by ovariectomies in mice, and therefore such inhibition has potential therapeutic value for osteoporosis [71,72]. The addition of 0.1 mM of ARL 67156 inhibited 8–13.7% of the total ATP hydrolysis in MVs, which indicated that TNAP was more effective in hydrolyzing ATP than the other ectoenzymes. 

### 2.3. Mineral Formation Monitored by Turbidity

For the determination of the influence of phosphatases other than GPI-anchored TNAP, a phosphatidylinositol phospholipase C (PI-PLC) treatment that can remove TNAP activity from MVs (66–92%) [73,74] was used. The use of PI-PLC resulted in the partial cleavage of TNAP and generated cTNAP (TNAP released from MVs) and cMVs (MVs without GPI-anchored TNAP). The integrity of the vesicles was preserved, although a minor increase in the diameter was observed in cMVs (286 ± 61 nm) compared to native MVs (217 ± 23 nm) after the PI-PLC treatment and ultracentrifugation process. The percentage of TNAP activity removed was around 66%, which was consistent with previous reports in the literature [75] (Table 2).

**Table 2 ijms-23-15072-t002:** Biochemical and biophysical characterization of MV; cMV and cTNAP obtained after treatment with PI-PLC as described in Materials and Methods. MV were treated with PI-PLC during 3 h, ultracentrifuged at 80,000× *g*. The supernatant contained cTNAP, and the pellet composed of cMV was resuspended in SCL without phosphate. Protein determination was assayed by the Bradford method [76]. TNAP activity in MVs isolated from chondrocytes, was measured continuously using para-nitrophenyl phosphate (pNPP, ε = 17,600 M^−1^·cm^−1^, pH 13.0, 1.0 M) [75]. Vesicles sizes were determined by Dynamic light scattering (DLS) at 25 °C using the unimodal distribution.

Sample	Protein (mg·mL^−1^)	pNPP (U·mg^−1^)	Diameter (nm)	PI
MV (native)	3.7 ± 0.3	4.4 ± 0.9	217 ± 23	0.4
cMV (partial cleaved TNAP)	1.9 ± 0.3	1.5 ± 0.3	286 ± 61	0.5
cTNAP (cleaved from MV)	0.4 ± 0.04	5.1 ± 1.8	--	--

SBI-425 inhibited a maximum of 33.4% of the total activity of the cTNAP, removed from the MV. This value is in agreement with the results of the experiments carried out previously with ATP at 3.0 mM as a substrate (Table 1) but less than expected based on the inhibition properties of human and mouse TNAP. The active site pocket of chicken TNAP has a Glutamine at position 434, rather than a Histidine present in human and mouse TNAPs, and this substitution explains the lower sensitivity of chicken TNAP to inhibition by SBI-425, as well as Levamisole [77]. In this sense, the inhibitory effect is not observed in MV since it has a high content of TNAP, while in cMV, its presence only delays mineralization (Figure 3).

In order to evaluate if NKA could affect mineralization, turbidity assays were performed to monitor the formation of calcium–phosphate precipitation by measuring the absorbance at 340 nm. ATP, as PP_i_, is a strong inhibitor of mineral formation and can delay the induction time of mineralization [74,78,79,80]. The addition of 3.42 mM of ATP delayed the starting induction time to around 20 h due to the time needed to hydrolyze ATP and to initiate mineralization (Figure 3A, control black symbols). 

The control experiment (curve without inhibitor) revealed an increase in turbidity with a sigmoidal shape. It started (starting time, t_i_) after 20 h of incubation and ended at 50.1 Abs_340nm_·mg^−1^ (U_max_) at the end of the 120 h of the experiment (t_f_ of Table 3). The addition of Levamisole (which inhibits TNAP-mediated ATP hydrolysis) almost eliminated the induction of mineralization due to the presence of excess ATP (conserved by the lack of ATPase activity) and, mostly, the absence of P_i_ (Figure 3A, red). The addition of ouabain with or without SBI-425 (Figure 3A, violet and green symbols) and SBI-425 alone (Figure 3A, cyan) slightly affected the starting induction time, which remained around 14–20 h (Table 3). However, the hydrolysis of ATP was not completely inhibited by ouabain and/or SBI-425 as compared to the control, since their t_max_ rate (58.8 to 74.4 h), U_max_ (Abs_340nm_·mg^−1^) (54.6–71.6) PMP (0.73–1.15 h^−1^) were almost similar and even slightly higher compared to the control values (t_max_ rate= 57.9 ± 0.8 h; U_max_ (Abs_340nm_·mg^−1^) = 50.1 ± 0.8 and PMP = 0.86 h^−1^). This indicated that an amount of ATP was sufficiently hydrolyzed even in the presence of ouabain and/or SBI-425.

The binding site pocket for the uncompetitive inhibitor SBI-425 is not perfectly conserved for chicken TNAP, as compared to that in human or mouse TNAP, due to the functionally significant H434Q substitution [77]. Furthermore, ATPase activity assays in Table 1 showed that 10 μM SBI-425 inhibited about 35% of MV total activity, whereas 5 mM Levamisole inhibited 70%. In this sense, the inhibitory effect cannot be observed in MV since there is a high abundance of TNAP that cannot be totally inhibited by SBI-425. Other phosphatases act as a backup mechanism for P_i_ production, in a manner analogous to the documented phosphatase activity of ENPP1 rather than its more traditional pyrophosphohydrolase activity, in the complete absence of TNAP activity [46].

Levamisole efficiently prevented mineral propagation. The inhibition is attributed mostly to TNAP. Chicken MVs do not have the same susceptibility to SBI-425 as for Levamisole since it appears that SBI-425 does not completely inhibit TNAP from chicken embryos, probably due to a distinct topology of the inhibition site of TNAP from chicken embryos as compared to that of human TNAP. Ouabain did not show a great effect on mineralization since the contribution of NKA compared to the hydrolysis of TNAP by MV was around 5% or less, and/or the ATP binding site of NKA would be mostly occluded inside of the vesicle. In order to verify the main role of TNAP in hydrolyzing ATP, the ability to hydrolyze ATP in cMVs was depleted by around 65% of TNAP, as compared to native MV. In Figure 3B, all mineralization curves associated with cMVs presented a delay of about 40 h (t_i_ in Table 3) when compared directly with the native MVs, but they retained the sigmoidal behavior. 

The delay corresponds to the step in the reaction, where ATP substrate and its derivatives were being hydrolyzed; however, there is still not enough P_i_ concentration for mineral formation. The addition of Levamisole completely abolished the mineralization. Even after removing 65% of TNAP (Figure 3B, Table 3), it was still impossible to observe any ouabain effect on the propagation and mineralization. The combined effect of Levamisole and ouabain could be explained in native MVs (Figure 3A) by the effect of Levamisole on TNAP activity since ouabain does not reach the binding site of NKA. The little increment in turbidimetry, when compared with Levamisole alone, could be explained by unspecific interaction and reducing the effect of Levamisole. In the cMV (Figure 3B), this effect was not observed since the contribution of TNAP activity was reduced by 70%.

### 2.4. Na,K-ATPase Proteoliposomes as MV Biomimetic Models

Due to the presence of several types of phosphatases in MVs and to the occluded NKA active site (not exposed to the external medium), it was unclear if NKA itself could contribute to P_i_ production inside MVs and to subsequent mineralization. In order to address this question, DPPC:DPPE proteoliposomes were constructed containing solubilized NKA from membrane fractions prepared from rabbit kidneys [62]. The concentration found for the solubilized NKA was 0.19 mg·mL^−1^, and its catalytic activity was 0.336 U·mg^−1^ (Table 4). 

The hydrodynamic diameter of the enzyme is 19 ± 1 nm (Table 4), obtained by the DLS analysis [64]. The crystal structures of the αβ and (αβ)_2_ conformations have a hydrodynamic radius of 5.0 and 6.3 nm (according to PDBs 3WGV and 3A3Y for the monomer and 3WGV for the dimer). Therefore, there is a balance between monomers, oligomers, and aggregates in solution as determined by analytical centrifugation [64].

Taking into account the mean diameter of the MVs and that of NKA, it is possible to determine the spatial contribution of the enzyme. The NKA diameters comprise 10, 5, and 6.3% of the mean diameter of the MVs when calculated by DLS or by crystallography, respectively. However, when these values are evaluated in surface area, 1134 nm^2^ is determined by the average found by DLS, 314 nm^2^ for the αβ form, 499 nm^2^ for the (αβ)_2_ form against 125,667 nm^2^ for the MVs with 200 nm of diameter. Therefore, in these calculations, the contribution of NKA to the total area of MVs varies from 0.9 to 0.25% of the area of MVs, which explains the difficulty in finding them during topographic analysis by AFM [66,81]. 

Mineral propagation parameters for TNAP and NPP1 liposomes in different lipid compositions indicated that the best results were obtained for the DPPC compositions [41,82], consistent with the presence of 41.8% phosphatidylcholine relative to the total phospholipid found in MVs [6]. The second largest phospholipid type is phosphatidylethanolamine which amounted to 14.9% of the total phospholipid composition in MVs [6]. The lipid composition can influence the catalytic activities of the enzymatic machinery. Another advantage of DPPE is that it induces an inside-out orientation, with the NKA active site facing the outside of the vesicles [63]. In this configuration, the active site of the enzyme is in an inside-out position and has free access to ATP from the medium [63,66]; moreover, as indicated by the similar activity values for the NKA (0.336 U·mg^−1^) and NKA-DPPC:DPPE (0.29 U·mg^−1^, corresponding to 86% of NKA activity) in contrast to NKA:DPPC (0.141 U·mg^−1^, corresponding to 61% of NKA activity), consistent with previous findings [63,66] (Table 4).

The mean diameter of DPPC liposomes measured by DLS was 365 ± 35 nm, while its polydispersity index (PI) was 0.7 (Table 4). The DPPC:DPPE liposome mean diameter was 504 ± 50 nm with a PI of 1.3 (Table 4). Such size discrepancies may reflect the nature of phospholipids and their respective PI values [66,83]. The NKA-DPPC has a mean diameter of 634 ± 60 nm, while the NKA-DPPC:DPPE has an average diameter of 899 ± 90 nm (Table 4). In order to induce mineral formation, a phosphtidylserine complex containing calcium and phosphate (PS-CPLX) was added since the enzyme alone was not able to nucleate. The solubilized protein, by hydrolyzing ATP (51.4 Abs_340nm_·mg^−1^), produced turbidity in the presence of PS-CPLX nucleators and Ca^2+^ ions, which amounted to 51.4 Abs_340nm_·mg^−1^ and had the same propagation capacity as NKA-DPPC:DPPE which was around 55.8 Abs_340nm_·mg^−1^ (Table 4). DPPC-NKA liposome by hydrolyzing ATP in a medium containing CPLX nucleators and Ca^2+^ ions induced a smaller nucleation profile which was 38.4 Abs_340nm_·mg^−1^ than the NKA-DPPC:DPPE. This is due to the more favorable orientation of the active site of NKA facing outside in NKA-DPPC:DPPE (90% inside-out orientation) as compared to that of NKA:DPPC (61%). Larger amounts of ATP are hydrolyzed by NKA-DPPC:DPPE than by NKA-DPPC. Other factors, such as interactions between the phospholipid DPPE and the DPPS used in the preparation of the PS-CPLX nucleator, cannot be completely ruled out. The inhibition of NKA by ouabain in proteoliposomes and in this assay condition has been extensively discussed by our group [62,63,84,85,86,87]. Nevertheless, the addition of ouabain did not completely prevent the ATP nucleation process, probably due to the occluded ouabain binding site, which turned to the lumen of proteoliposomes, but even the addition of detergent was not able to improve the inhibition and abrogate mineral formation. 

### 2.5. Spectroscopic Analysis of Minerals Formed by Na, K-ATPase-Liposomes

The ATP-FTIR spectra of the minerals formed from the incubation of NKA-DPPC (Figure 4A1–A3) and of NKA-DPPC:DPPE (Figure 4B1–B3) have similar profiles. The remaining activity of the NKA-liposomes after the inhibition by ouabain did not affect the mineral quality, even after the addition of C_12_E_8_ to solubilize the vesicles. That is, the residual activity was sufficient to generate phosphate and mineral formation. The 1040 cm^−1^ band (as indicated by the dashed yellow line) could correspond to apatite, suggesting that the proteoliposome formed by NKA and either DPPC or DPPE:DPPE could promote mineral formation. 

## 3. Materials and Methods

### 3.1. Materials

Solutions were prepared with Millipore Direct-Q ultrapure apyrogenic water. All the reagents were of the highest commercial purity available. Bovine serum albumin (BSA), Collagenase type 1 (*clostridium histolyticum*), Tris[hydroxymethyl]aminomethane (Tris), N-(2-hydroxyethyl) piperazine-N’-ethanesulfonic acid (HEPES), adenosine 5′-Triphosphate tris salt (ATP), dodecyloctaglycol (C_12_E_8_), Glutaraldehyde (Grade I, specially purified for use as an electron microscopy fixative), Octylphenoxy poly(ethyleneoxy)ethanol, (branched), p-Nitrophenyl phosphate disodium hexahydrate, Phospholipase-C (*bacillus cereus*), Levamisole (L(−)-2,3,5,6-Tetrahydro-6- phenylimidazo [2,1-*b*]thiazole hydrochloride), ouabain (1β,3β,5β,11α,14,19-Hexahydroxycard- 20(22)-enolide 3-(6-deoxy-α-L-mannopyranoside)), ARL-67156 trisodium salt, a selective inhibitor of ecto-ATPase, D-glucose and sucrose were purchased from Sigma Chemical Co. (St. Louis, MO). 1,2-dipalmitoyl-sn-glycero-3-phosphocholine (DPPC) and 1,2-dipalmitoyl-sn-glycero-3-phosphoethanolamine (DPPE) were acquired from Avanti Polar Lipids. Ethylenediaminetetracetic acid (EDTA), magnesium chloride (MgCl_2_), potassium chloride (KCl), sodium chloride (NaCl), sodium hydrogen carbonate (NaHCO_3_), di-sodium sulfate (Na_2_SO_4_) and trichloroacetic acid (TCA) were from Merck. Biobeads and Bradford reagents were acquired from Bio-Rad. The TNAP inhibitor SBI-425 was kindly provided by Professor Dr. J.L. Milan of the Sanford Health Institute in La Jolla, California.

### 3.2. Matrix Vesicles Isolated from Chicken Embryo Femurs

The MVs were prepared with a collagenase digestion step [75]. Briefly, 20 chicken embryos (17 days after fertilization) were sacrificed by decapitation. The femurs were dissected, and slices (1–3 mm thick) of the epiphyses/growth plates finely cut and digested for 3 h, at 37 °C, in synthetic cartilage buffer (SCL) buffer supplemented with 1 mM CaCl_2_ and type I collagenase, concentration of 300 U/g of tissue. SCL, a buffer that mimics the native environment of MVs in the cartilage, was composed of 1.83 mM NaHCO_3_, 12.7 mM KCl, 0.57 mM MgCl_2_, 5.55 mM D-glucose, 63.5 mM sucrose, 16.5 mM Tris (2-Amino-2-hydroxymethyl-propane-1,3-diol), 100 mM NaCl, 0.7 mM Na_2_SO_4_ in water at pH 7.6 [31]. After digestion, the suspension was filtered through a nylon membrane (100 μm) and centrifuged at 600× *g* to remove all cell debris. The supernatant was subjected to consecutive centrifugations at 20,000× *g* for 30 min and 80,000× *g* for 1 h, both at 4 °C. The final pellet was homogenized in 200 μL of SCL with 2 mM NaCl_2_ and stored at 4 °C. Protein determination was assayed with 2 uL of MV suspension and add 798 μL pure water, and 200 μL Bradford [76]. All procedures involved in the euthanasia of animal embryos were approved by the ethics committee of FFCLRP protocol 19.1.842.59.13.

### 3.3. TNAP Enzymatically Cleaved from MV by PI-PLC

Aliquots of native MV were incubated with phosphatidylcholine phospholipase C (PI-PLC) (0.1 U) in SCL buffer for 3 h under constant rotating agitation at 37 °C [73]. Samples were further centrifuged at 80,000× *g* for 1 h at 4 °C. The supernatant containing TNAP cleaved from its GPI anchor was reserved. The pellet containing MV with partial cleated TNAP (cMV) was resuspended in the same volume of SCL buffer. Both samples had protein concentration and specific activity quantified by [76] and [78], respectively. The size of cMV was analyzed by DLS.

### 3.4. Mineralization Assay for MV and for cMV

Mineralization for MV and cMV samples were performed without phosphatidylserine-calcium complex phosphatidylserine calcium complex nucleator (PS-CPLX): MV and cMV were incubated in 96 wells plates with SCL for 120 h at 37 °C in a hydrated incubation chamber. The SCL medium was supplemented with the inhibitors Levamisole 10.0 mM, SBI-425 μM, and ouabain 3.0 mM either alone or in combination as indicated. Samples were pre-incubated for 30 min before 3.41 mM ATP addition. The absorbance at 340 nm was measured within 24-h intervals and normalized by protein concentration generating Abs_340nm_/mg data. Curves of Abs_340nm_/mg versus time were plotted, and the sigmoidal tendency of mineral formation was evaluated by the mathematical approach described [88]. Mineralization-related parameters obtained were: the initial mineralization time (t_i_) is characterized by a rapid increase in U; the final mineralization time (t_f_) is characterized by a decrease in U; the time in which the maximum rate of mineral formation is reached (t_max rate_) corresponds to the maximum of the dU/dt curve; U_max_ is the maximum of turbidity (Abs_340nm_/mg); U_max_/t_max rate_ is the potential of mineral propagation (PMP) that is a measure of the tendency to form mineral [88]. Data shown in Table 3.

### 3.5. Preparation of Na,K-ATPase

Solubilized NKA was obtained from dark red medulla of rabbit kidney as described [62]. Protein concentration was estimated in the presence of 2% SDS (0.2 g/mL) [63], and BSA was used as standard. ATPase activity was assayed discontinuously at 37 °C by quantifying phosphate release using standard conditions: 50.0 mM HEPES buffer, pH 7.5, containing 3.0 mM ATP, 10.0 mM KCl, 5.0 mM MgCl_2_ and 50.0 mM NaCl in a final volume of 1.0 mL. The reaction was initiated by addition of the enzyme, stopped with 0.5 mL of cold 30% TCA at appropriate time intervals, followed by 4000× *g* centrifugation immediately prior to phosphate measurement as described [62,63,69]. All procedures involved in the euthanasia of animals were approved by the ethics committee of FFCLRP protocol 17.5.801.59.9.

### 3.6. Liposomes and Na,K-ATPase Proteoliposomes

DPPC and 1:1 DPPC:DPPE (*w*/*w*) multilayer liposomes (1.0 mg/mL) were sonicated during 1 min per milliliter of the sample at 200 W (VibraCell, VC-600 with a microtip) to obtain large unillamelar vesicles (LUVs) [73]. DPPC and 1:1 DPPC:DPPE (*w*/*w*) NKA-liposomes were prepared by co-solubilization as previously described [63,66]. Vesicle size was determined by Dynamic light scattering (DLS) at 25 °C using the unimodal distribution. Determination of the protein incorporation in proteoliposomes was performed according to method previously described by Cornelius et al. [61], and BSA was used as standard.

### 3.7. Mineralization Assay for Na,K-ATPase Proteoliposomes

Mineralization was assayed with a nucleator PS-CPLX. Synthesis of PS-CPLX was performed as previously described [41,82,88,89]. Solubilized NKA, NKA-liposomes (NKA-DPPC and NKA-DPPC:DPPE), and MV were incubated with PS-CPLX in the 96 wells plates for 48 h in synthetic cartilage lymph (SCL) buffer with PS-CPLX. To start the reaction 3.41 mM of ATP was added to the medium. Only initial and end-point were measured to observe the total contribution of NKA. Absorbance was measured at 340 nm and normalized by protein concentration in the assay resulting in Abs_340nm_·mg^−1^ information. Abs_340nm_·mg^−1^ = [Δ(Abs_340nm_)/(NKA per well)] was used to compare the ability of mineralization per mg of enzyme used in the assay. Experiments were performed in triplets and are presented in Table 4. Mineralization was also assayed for NKA-liposomes and MV without PS-CPLX in the same conditions described above, but using 3.0 mM ouabain and/or 0.2% C_12_E_8_, always with pre-incubated for 30 min before ATP addition.

### 3.8. Atomic Force Microscopy

MV samples were prepared with SCL buffer. Samples were stabilized by adding 1:10 (*v/v*) of glutaraldehyde solution (25%), and 5 μL of the sample was placed onto freshly cleaved mica substrate. After stabilization, samples were imaged as described [9]. AFM micrographs were obtained by Shimadzu SPM-9600 Scanning Probe Microscopy (Shimadzu Corporation, Kyoto, Japan) operating in tapping mode (called Phase Contrast Mode by Shimadzu), which is a dynamic-based mode. Scanning was performed in air at 25 °C by using standard 4-sided pyramidal silicon probes with a resonance frequency ranging from 324 to 369 kHz (Nanosensors™, Neuchatel, Switzerland). The scan rate was set at 0.2–0.3 Hz to prevent tip-induced vesicle deformations and/or damages. The values of the spring constants of the cantilevers were approximately 38 ± 8 N/m. Tips radius was 7 nm (guaranteed by the manufacturer >10 nm) and 10–15 μm height. Their resonance frequency values were approximately 336 ± 67 kHz. The vesicle sizes were determined by SPM Offline software from Shimadzu. For each sample, N = 100 vesicles were analyzed.

### 3.9. Determination of ATP Hydrolysis by Colorimetric Assay and by ^31^P NMR

ATPase activity was colorimetrically determined discontinuously at 37 °C, and ^31^P NMR activity was determined by measuring inorganic phosphate released at 25 °C in 50.0 mM HEPES reaction medium, pH 7.5, containing 3.0 mM ATP, 10.0 mM KCl, 5.0 mM MgCl_2_ and 50.0 mM NaCl and 10% ^2^H_2_O for ^31^P NMR assay [62,63,69]. Evolution 60S UV-Visible Spectrophotometer (Thermo Scientific) spectrophotometer was used for colorimetry. ^31^P NMR spectra were measured with a 300 ultrashied Bruker spectrometer. One unit of enzyme (1 U) was arbitrarily defined as 1 μmol of phosphate released per minute under standard test conditions (when specified, U/mg was used). Results represent the mean of determinations performed in three independent measurements.

### 3.10. Determination of p-Nitrophenolphosphate Hydrolysis by Colorimetric Assay

TNAP activity in MVs isolated from chondrocytes, was measured continuously using para-nitrophenyl phosphate (pNPP, ε = 17,600 M^−1^·cm^−1^, pH 13.0, 1.0 M) [75]. Aliquots of MVs were mixed with 200 μL of 10 mM pNPP dissolved in 56 mM 2-Amino-2-methyl-1-propanol buffer (Ampol) pH = 10.0 and 2 mM MgCl_2_ at 37 °C. The absorbance was recorded at 410 nm for 120 s using a 96-well plate reader. Three independent measurements were performed. TNAP activity was calculated as the number of units (U) per mg of total protein (1 U corresponds to 1 μmol of hydrolyzed pNPP per minute). Alternatively, TNAP activity measurement was performed discontinuously. For these experiments, when at optimal pH, 10 mM pNPP dissolved in Ampol 50 mM pH 10.0, 2 mM MgCl_2_ buffer was used at 37 °C in a final volume of 0.5 mL. The reaction was started by adding an aliquot of the samples and stopped by adding 0.5 mL of 1.0 M NaOH at appropriate time intervals [90,91]. When specified, U/mg was used.

### 3.11. FTIR Chemical Analysis of Mineral Formed

The chemical composition of the dried precipitates from the mineralization assays was investigated by means of Fourier-transformed infrared spectroscopy using an attenuated total reflectance accessory (ATR-FTIR, germanium crystal with acquisition from 4000–600 cm^−1^) model IRPrestige-21, Shimadzu Co., Tokyo, Japan.

### 3.12. Statistical Analysis

Kinetics and mineralization data are reported as the mean ± SD of triplicate measurements of three different proteoliposome preparations, which was considered to be statistically significant at *p* ≤ 0.05 or *p* ≤ 0.001, as indicated with “*” in the tables.

## 4. Conclusions

Extracellular ATP is hydrolyzed mostly by TNAP in MVs, since 5 mM Levamisole inhibited around 69–72% of the total ATP hydrolysis, while around 8–13% was hydrolyzed by other phosphatases, including NTPDase1, NTPDase3, and NPP1 as revealed by the addition of 0.1 mM ARL which can inhibit these phosphatases. TNAP’s main function in MVs is to hydrolyze PP_i_, which is a strong inhibitor of apatite formation [80,92]. ATP and ADP can also inhibit apatite formation [93]. Therefore, TNAP is an essential enzyme to induce mineralization by depleting PP_i,_ ATP, and ADP [80,92]. To sustain internal apatite formation in the lumen of MVs, P_i_ has to be either transported into the MVs [26,27,28] and/or produced inside MVs (Figure 1). Several enzymes, including PHOSPHO1 [51,52] and ATPases [53,54], could contribute to the production of internal P_i_ (Figure 1). Here, NKA accounted for not more than 3–8% of the total ATP hydrolysis, consistent with other reports [53,67]. On the other hand, 164 mM Na^+^ and 19 mM K^+^ were found in nascent MVs in contrast to 59 mM Na^+^ and 101 mM K^+^ in hypertrophied chondrocytes, from which MVs originate [6]. This suggests that all the ATP was depleted inside MVs and that NKA in MVs could not fulfill its function to maintain ion asymmetry. 

The question remains whether NKA could induce apatite formation inside MVs with a sufficient amount of ATP, corresponding to the very early stage of mineralization, before the complete depletion of ATP. Since the ATP hydrolysis was dominated by TNAP and other phosphatases, liposomes were prepared to reconstitute NKA to address the question. Solubilized NKA, as well as NKA-DPPC and NKA-DPPC:DPPE liposomes, were able to propagate minerals in SCL buffer in 48 h in the presence of ATP, Ca^2+^, and the PS-CPLX nucleator. The NKA-DPPC:DPPE liposome induced greater mineralization, as compared to NKA-DPPC liposomes, because of the better orientation of the ATP active site, which was mostly oriented outside due to DPPE presence [63,66,87]. ATR-FTIR analysis of the mineral formed by NKA proteoliposomes indicated the presence of apatite. The findings support the contention that ATP could be a source of internal P_i_, promoting the formation of calcium phosphate complexes, including apatite. It is tempting to suggest that all ATP inside MVs is hydrolyzed by NKA, providing a part of the P_i_ necessary for apatite formation just after their release from hypertrophied chondrocytes, but MVs do not have the mitochondrial machinery to continuously supply ATP, which may cause the loss of Na^+^/K^+^ asymmetry in MVs [6]. Other internal enzymes, such as PHOSPHO1 [51,52], can continue to produce a sufficient amount of luminal P_i_ to sustain apatite formation. Thus, we propose the putative function of Na,K-ATPase is to transport Na^+^ out MVs, which could be used by the P_i_T-1 cotransporter in the initial stages of mineralization (sufficient for nucleation). The P_i_ produced by Na,K-ATPase from ATP could contribute to the pool of P_i_ produced by PHOSPHO1.

## Figures and Tables

**Figure 1 ijms-23-15072-f001:**
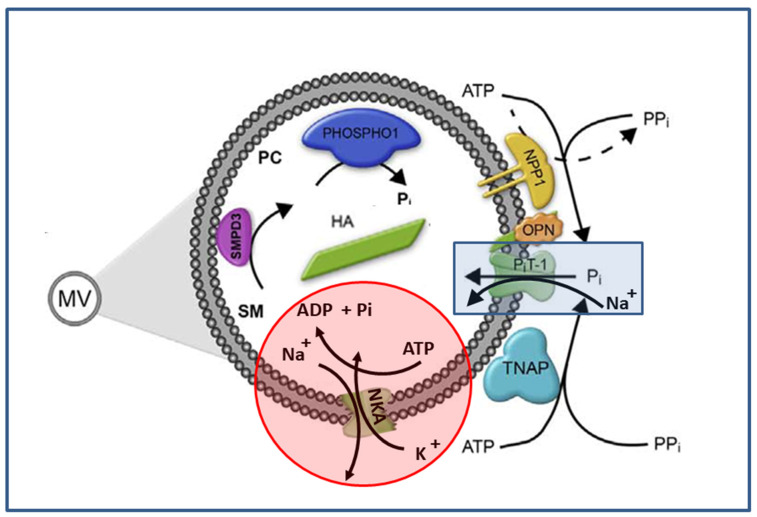
P_i_ homeostasis during MV-mediated mineralization, adapted from Reference [10]. MVs provide an environment that allows the initial nucleation of apatitic mineral inside the vesicles as well as to provide nucleation sites that can help propagate the apatitic crystals onto the extracellular matrix through interaction with the collagen fibers. Orphan phosphatase 1 (PHOSPHO1) produces P_i_ from the hydrolysis of phosphocholine (PC), which itself is derived from sphingomyelin (SM) by the action of sphingomyelin phosphodiesterase 3 (SMPD3) located in the inner surface of the MV membrane. In addition, phosphate transporter 1 (P_i_T-1, in blue box) and possibly other unidentified transporter(s) help supply P_i_ ions generated perivesicularly to help nucleation of apatite (HA) inside the MVs. TNAP and NPP1 both control the extracellular PP_i_/P_i_ ratio. An Na^+^ gradient could be generated by NKA (red circle) after ATP hydrolysis inside MVs, helping increase the P_i_ pool generated by PHOSPHO1.

**Figure 2 ijms-23-15072-f002:**
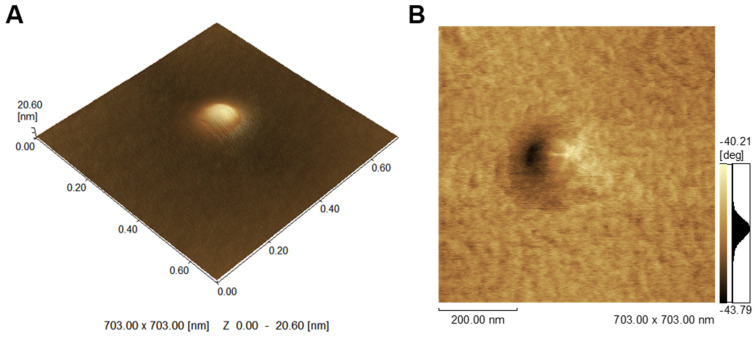
Tapping mode AFM images of isolated MV prepared in SCL buffer without phosphate and placed in fresh cleaved mica. (**A**) 3D topographic image showing the vesicle’s topology. (**B**) Phase image showing short range of phase shift in the vesicle’s topology, revealing a high protein complexity. Scanning was performed in air at 25 °C by using standard 4-sided pyramidal silicon probes with a resonance frequency ranging from 324 to 369 kHz (Nanosensors™, Switzerland). The scan rate was set at 0.2–0.3 Hz to prevent tip-induced vesicle deformations and/or damages. The vesicles sizes were determined by SPM Offline software, from Shimadzu. For each sample, N = 100 vesicles were analyzed.

**Figure 3 ijms-23-15072-f003:**
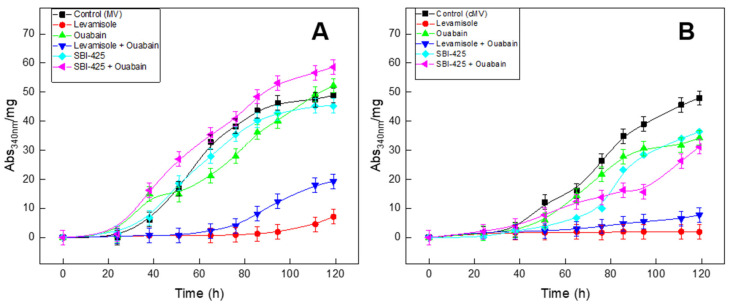
MV (**A**) and cMV (**B**) were incubated in 96 wells plates with SCL for 120 h at 37 °C in an incubation hydrated chamber. The SCL medium with 2.0 mM of CaCl_2_ was supplemented with the inhibitors Levamisole 10.0 mM, SBI-425 10 μM and Ouabain 3.0 mM either alone or in combination as indicated. Samples were pre-incubated for 30 min before the addition of 3.41 mM ATP. The absorbance at 340 nm was measured within 24 h intervals and normalized by protein concentration generating Abs_340nm_/mg data and resulting in the kinetic parameters presented on Table 3, as described in Materials and Methods. Control (absence of inhibitor) (■ black); Levamisole, phosphatase inhibitor (red ●); SBI-425, TNAP specific inhibitor (cyan ◆); Ouabain, NKA specific inhibitor (green ▲); Levamisole + Ouabain (▼ blue); SBI-425 + Ouabain (◄ pink).

**Figure 4 ijms-23-15072-f004:**
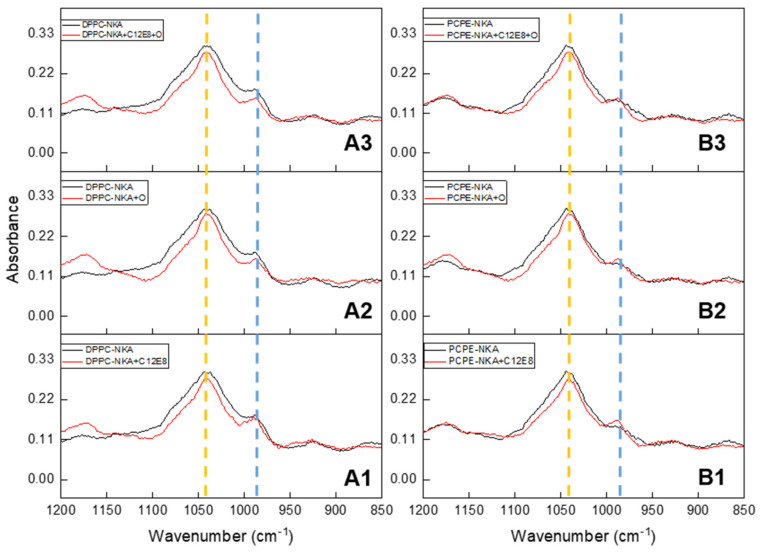
ATR-FTIR absorbance spectra of dehydrated minerals obtained from DPPC-NKA and DPPC:DPPE-NKA liposomes samples after 48 h’ incubation with ATP, 0.2% C_12_E_8_ detergent and 3.0 mM Ouabain (NKA inhibitor) when described. Bands of HPO_3_^2−^ (987 cm^−1^) and PO_4_^3−^ (1040 cm^−1^) are indicated in blue and orange dash lines, respectively. (**A1**–**A3**) DPPC-NKA liposomes. (**A1**) Effect of C_12_E_8_. (**A2**) Effect of Ouabain. (**A3**) Effect of combination of 0.2% C_12_E_8_ and 3 mM Ouabain. (**B1**–**B3**) DPPC-DPPE-NKA liposomes. (**B1**) Effect of 0.2% C_12_E_8_. (**B2**) Effect of Ouabain. (**B3**) Effect of combination of 0.2% C_12_E_8_ and 3 mM Ouabain. In the figures, Black line was used for the control spectra with only NKA-liposome, A1 to A3 DPPC-NKA and B1 to B3 DPPC:DPPE-NKA (▬) and red line represent the spectra with 3 mM Ouabain or 0.2% C_12_E_8_ (▬).

**Table 1 ijms-23-15072-t001:** Comparison of ATP hydrolysis activity determined by colorimetric and ^31^P NMR method. The activities were determined in % considering 100% activity in the absence of inhibitor under optimal conditions for the determination of NKA activity, as described in materials and Methods. Ouabain (NKA specific inhibitor); Levamisole and SBI-425 (TNAP inhibitors); ARL-67156 (CD39 specific inhibitor). ATPase activity was colorimetrically determined discontinuously, at 37 °C and ^31^P NMR activity was determined by measuring inorganic phosphate released, in 50.0 mM HEPES reaction medium, at 25 °C using medium with pH 7.5, containing 3.0 mM ATP, 10.0 mM KCl, 5.0 mM MgCl_2_ and 50.0 mM NaCl and 10% ^2^H_2_O for ^31^P NMR assay [62,63,69].

ATPase Activity Method	Ouabain (3 mM)	Levamisole (5 mM)	ARL (0.1 mM)	SBI-425 (5 μM)	SBI-425 (10 μM)
Colorimetric	93.2 ± 3.6	27.8	92.0 ± 5.6	78.5	64.6
^31^P NMR	97.6 ± 2.9	31.4 ± 3.9	86.3 ± 4.2	ND	ND

ND (Not determined).

**Table 3 ijms-23-15072-t003:** Kinetic parameters of nucleation process obtained from the mineralization curves for MV and cMV, after 120 h assay in SCL buffer supplemented with 2.0 mM of CaCl_2_, and inhibitors. Data are reported as the mean ± SD of triplicate measurements of three different MV preparations.

	Inhibitor (Concentration)	t_i_ (h)	t_f_ (h)	t_max rate_ (h)	U_max (Abs340nm·mg^−1^)_	PMP (h^−1^)
MV	Without	20.0 ± 0.4	120.0 ± 0.6	57.9 ± 0.8	50.1 ± 0.8	0.86 ± 0.04
	Levamisole (5 mM)	ND	ND	ND	ND	ND
	Ouabain (3 mM)	24.3 ± 0.7	160.5 ± 7.6	74.4 ± 0.7	54.6 ± 3.3	0.73 ± 0.03
	Levamisole (5mM) + Ouabain (3 mM)	60.0 ± 0.3	119.0 ± 1.6	91.7 ± 0.5	21.4 ± 0.8	0.23 ± 0.02
	SBI-425 (10 μM)	15.0 ± 0.4	120.0 ± 2.4	58.9 ± 0.3	48.8 ± 1.1	0.83 ± 0.05
	SBI-425 (10 μM) + Ouabain (3 mM)	14.6 ± 0.2	140.7 ± 1.5	62.3 ± 5.6	71.6 ± 2.3	1.15 ± 0.07
cMV	Without	34.1 ± 1.3	140.9 ± 5.2	79.7 ± 3.4	58.6 ± 3.7	0.74 ± 0.04
	Levamisole (5 mM)	ND	ND	ND	ND	ND
	Ouabain (3 mM)	35.9 ± 1.6	125.6 ± 4.1	35.2 ± 1.0	69.8 ± 1.2	1.98 ± 0.09
	Levamisole (5 mM) + Ouabain (3 mM)	ND	ND	ND	ND	ND
	SBI-425 (10 μM)	48.9 ± 1.8	128.2 ± 3.9	37.4 ± 2.3	83.3 ± 2.1	2.22 ± 0.11
	SBI-425 (10 μM) + Ouabain (3 mM)	37.6 ± 3.8	94.4 ± 2.4	15.6 ± 2.2	55.9 ± 4.8	3.58 ± 0.14

Mineralization-related parameters obtained were: the initial mineralization time (t_i_) is characterized by a rapid increase in U; the final mineralization time (t_f_) is characterized by a decrease in U; the time in which the maximum rate of mineral formation is reached (t_max rate_) corresponds to the maximum of the dU/dt curve; Umax is the maximum of turbidity (Abs_340nm/mg_); U_max/tmax_ rate is the potential of mineral propagation (PMP) that is a measure of the tendency to form mineral [32]. ND, not determined.

**Table 4 ijms-23-15072-t004:** Biochemical and biophysical characteristics of NKA solubilized and NKA-liposomes prepared in DPPC and DPPC:DPPE. Protein quantification was assayed by Cornelius et al. [65]. ATPase activity was assayed discontinuously at 37 °C by quantifying phosphate release using standard conditions: 50.0 mM HEPES buffer, pH 7.5, containing 3.0 mM ATP, 10.0 mM KCl, 5.0 mM MgCl_2_ and 50.0 mM NaCl as described in [62,63,69]. The ability of mineralization was tested in SCL buffer with 2.0 mM of CaCl_2_, in the presence of PS-CPLX nucleator and 48 h of incubation at 37 °C with 3.41 mM ATP as Pi source. Absorbance was measured at 340 nm and normalized by protein concentration in the assay resulting in Abs_340nm_·mg^−1^.

Sample	NKA (mg·mL^−1^)	ATPase (U·mg^−1^)	Diameter (nm)	IP	Abs_340nm_·mg^−1^
NKA solubilized	0.19	0.336	19.1 ± 1 *	--	51.9 ± 5
DPPC	---	---	365 ± 35	0.7	---
DPPC:DPPE	---	---	504 ± 50	1.3	---
NKA-DPPC	0.61	0.141	634 ± 60	1.4	38.4 ± 4
NKA-DPPC:DPPE	0.53	0.290	899 ± 90	1.4	51.4 ± 5

* Values from reference [65].

## Data Availability

Not applicable.

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
