# Peer review of "Shedding Light on the Role of Na,K-ATPase as a Phosphatase during Matrix-Vesicle-Mediated Mineralization†"

_ijms, 2022, doi:10.3390/ijms232315072_

Round 1
Reviewer 1 Report
Endochondral ossification is a carefully orchestrated process mediated by promoters and inhibitors of mineralization. Phosphatases are long implicated, but their identities and functions remain unclear. Alkaline phosphatase (TNAP) plays a crucial role in promoting the mineralization of the extracellular matrix by restricting the concentration of the calcification inhibitor inorganic pyrophosphate (PPi). PHOSPHO1, a phosphatase with specificity for phosphoethanolamine and phosphocholine, plays a functional role in the initiation of calcification, and the ablation of PHOSPHO1 and TNAP function prevents skeletal mineralization. PHOSPHO1 is involved in the first step of MV-mediated initiation of mineralization during endochondral ossification.
MVs do not have the mitochondrial machinery to supply ATP continuously, which may cause the loss of Na+/K+ asymmetry in MVs. As noted by the authors, other internal enzymes, such as PHOSPHO1, continue to produce a sufficient amount of luminal Pi to sustain the apatite formation. PHOSPHO1 has a nonredundant functional role during endochondral ossification, and based on these data and a review of the current literature, an inclusive model involving intravesicular PHOSPHO1 function and the Pi influx into MVs during the initiation of mineralization was proposed; the functions of TNAP, nucleotide pyrophosphatase phosphodiesterase-1, and collagen in the extravesicular progression of mineralization was supported. Current manuscript completely ignored the contribution of PHOSPHO1 in accounting for MV mineralization.
It is well established that the Na, K-ATPase enzyme complex consists of two polypeptide chains: the α-subunit or catalytic chain, which is responsible for the enzymatic activity of the complex, and the β-subunit, which has a structural function and may also play a functional role in the catalytic reaction and ion-pumping mechanism. A third γ-subunit is a small hydrophobic proteolipid, which is associated with the Na, K-ATPase, which, although not essential either for catalytic activity or ion transport, acts as a kinetic regulator.
The α- and β-subunits also present additional thermal stability that may be modulated by the nature of the co-solvent (detergent or lipid) used in the preparations of the Na, K-ATPase. In addition, distinct processes of β-subunit displacement and α-α-subunit aggregate formation may also contribute to the changes in the CD spectra and the enzyme activity. The authors have not considered how reconstituted enzyme use differs from the detergent-solubilized enzyme.
Other studies have shown pH dependence of sodium-independent ouabain-sensitive ATP hydrolysis. The optimal range of ouabain dose is 5-10 mM. However, there was a huge difference when used at 2 and 5 mM but not at three mM concentrations. How would the authors explain the discrepancy? Was pH ever considered in the hydrolysis experiments?
Minor issues:
The results section has too much background information.
Legends should contain sufficient detail to make the figure easily understood. It is desirable to have expanded legends/footnotes for figures.
Figure 1: Incorrect reference/typo. AFM analysis in Figure 2.
The abbreviation cMV needs to be clarified. “The cleavage resulted in cTNAP (hereinafter referred to as TNAP cleaved from 219 MVs) and cMVs (TNAP free MVs).” It would be appropriate to use the acronym cMV for cleaved MVs rather than TNAP-free MVs.
Conclusions: “Since five mM levamisole, 356 inhibited around 72-69 % of the total ATP hydrolysis,” pl. change this to 69-72%.
Suggestion: Consider restructuring the paragraph - Premise first and findings later.
Author Response
Referee #1
Endochondral ossification is a carefully orchestrated process mediated by promoters and inhibitors of mineralization. Phosphatases are long implicated, but their identities and functions remain unclear. Alkaline phosphatase (TNAP) plays a crucial role in promoting the mineralization of the extracellular matrix by restricting the concentration of the calcification inhibitor inorganic pyrophosphate (PPi). PHOSPHO1, a phosphatase with specificity for phosphoethanolamine and phosphocholine, plays a functional role in the initiation of calcification, and the ablation of PHOSPHO1 and TNAP function prevents skeletal mineralization. PHOSPHO1 is involved in the first step of MV-mediated initiation of mineralization during endochondral ossification.
ANSWSER: thank you for your comments we elaborated a small text about PHOSPHO1 in the introduction, but we haven’t neglected this enzyme, it is presented in Figure 1 and commented in the text (Page 5 of introduction). Now we included more details about this enzyme in the introduction. Please see revised version.
MVs accumulate in the lumen the Pi generated by specific enzymes. PHOSPHO1 is one of most prominent example of this enzymes. Its hydrolysis substrates are phosphocholine and phosphoethanolamine produced locally by activity of the neutral sphingomyelinase 2 (encoded by sphingomyelin phosphodiesterase 3 gene SMPD3) [1-3]. In addition, Pi is also generated extracellularly by tissue-nonspecific alkaline phosphatase (TNAP [4]) from the hydrolysis of phosphoesters as well as by ectonucleotide pyrophosphatase/phosphodiesterase 1 (NPP1) from the hydrolysis of extracellular ATP [5] and incorporated in the MVs′ lumen via specific sodium-dependent (e.g., PiT-1) [6–8] or sodium-independent [9] transporters. How MVs accumulate calcium in their lumen has not been fully described. Nevertheless, the gradient of Na+ generated by NKA along with its own ATP hydrolysis could be theoretically sufficient to produce a burst of Pi inside of the MVs to form the nucleational core, using the auxiliary action of PiT1 cotransporter (Figure 1).
- Stewart, A.J.; Roberts, S.J.; Seawright, E.; Davey, M.G.; Fleming, R.H.; Farquharson, C. The Presence of PHOSPHO1 in Matrix Vesicles and Its Developmental Expression Prior to Skeletal Mineralization. Bone 2006, 39, 1000–1007. https://doi.org/10.1016/j.bone.2006.05.014.
- Dillon, S.; Staines, K.A.; Millán, J.L.; Farquharson, C. How To Build a Bone: PHOSPHO1, Biomineralization, and Beyond. JBMR Plus 2019, 3, e10202. https://doi.org/10.1002/jbm4.10202.
- Thouverey, C.; Malinowska, A.; Balcerzak, M.; Strzelecka-Kiliszek, A.; Buchet, R.; Dadlez, M.; Pikula, S. Proteomic Characterization of Biogenesis and Functions of Matrix Vesicles Released from Mineralizing Human Osteoblast-like Cells. J. Proteom. 2011, 74, 1123–1134. https://doi.org/10.1016/j.jprot.2011.04.005.
- Majeska, R.J.; Wuthier, R.E. Studies on Matrix Vesicles Isolated from Chick Epiphyseal Cartilage Association of Pyrophosphatase and ATPase Activities with Alkaline Phosphatase. Biochim. Biophys. Acta (BBA) Enzymol. 1975, 391, 51–60. https://doi.org/10.1016/0005-2744(75)90151-5.
- Hessle, L.; Johnson, K.A.; Anderson, H.C.; Narisawa, S.; Sali, A.; Goding, J.W.; Terkeltaub, R.; Millán, J.L. Tissue-Nonspecific Alkaline Phosphatase and Plasma Cell Membrane Glycoprotein-1 Are Central Antagonistic Regulators of Bone Mineralization. Proc. Natl. Acad. Sci. USA 2002, 99, 9445–9449. https://doi.org/10.1073/pnas.142063399.
- Montessuit, C.; Bonjour, J.P.; Caverzasio, J. Expression and Regulation of Na‐Dependent Pi Transport in Matrix Vesicles Produced by Osteoblast‐like Cells. J. Bone Miner. Res. 1995, 10, 625–631. https://doi.org/10.1002/jbmr.5650100416.
- Guicheux, J.; Palmer, G.; Shukunami, C.; Hiraki, Y.; Bonjour, J.P.; Caverzasio, J. A Novel in Vitro Culture System for Analysis of Functional Role of Phosphate Transport in Endochondral Ossification. Bone 2000, 27, 69–74. https://doi.org/10.1016/S8756-3282(00)00302-1.
- Yadav, M.C.; Bottini, M.; Cory, E.; Bhattacharya, K.; Kuss, P.; Narisawa, S.; Sah, R.L.; Beck, L.; Fadeel, B.; Farquharson, C.; et al. Skeletal Mineralization Deficits and Impaired Biogenesis and Function of Chondrocyte-Derived Matrix Vesicles in Phospho1-/- and Phospho1/Pit1 Double-Knockout Mice. J. Bone Miner. Res. 2016, 31, 1275–1286. https://doi.org/10.1002/jbmr.2790.
- Solomon, D.H.; Browning, J.A.; Wilkins, R.J. Inorganic Phosphate Transport in Matrix Vesicles from Bovine Articular Cartilage. Acta Physiol. 2007, 190, 119–125. https://doi.org/10.1111/j.1748-1716.2007.01670.x.
MVs do not have the mitochondrial machinery to supply ATP continuously, which may cause the loss of Na+/K+ asymmetry in MVs. As noted by the authors, other internal enzymes, such as PHOSPHO1, continue to produce a sufficient amount of luminal Pi to sustain the apatite formation. PHOSPHO1 has a nonredundant functional role during endochondral ossification, and based on these data and a review of the current literature, an inclusive model involving intravesicular PHOSPHO1 function and the Pi influx into MVs during the initiation of mineralization was proposed; the functions of TNAP, nucleotide pyrophosphatase phosphodiesterase-1, and collagen in the extravesicular progression of mineralization was supported. Current manuscript completely ignored the contribution of PHOSPHO1 in accounting for MV mineralization.
ANSWSER: Thank you for your comments, the PHOSPHO1 contribution in Pi generation inside of MVs was considerate even more important than NKA, our hypothesis is that Na,K-ATPase could have a complementary role for the amount Pi to start the formation of the nucleational core. We agree and share the referee’s argument that MVs does not have a mitochondrial machinery to supply ATP continuously. The mineralization process supported by Na,K-ATPase would not be a viable hypothesis. The function of this protein is not related to the total contribution in the Pi production. In the time line of the MVs lifetime, we have placed our putative suggestion to occupy the initial stage of mineralization, enhancing the concentration of Pi inside of MVs using ATP hydrolysis and the cotransport of Na+ by phosphate transporter 1 (PiT-1). We highlight that the above mentioned step should be considered as an additional secondary step to the main step, which is the Pi produced by PHOSPHO1 generates mineral nucleation, forming apatite (HA) See legend of Figure 1. So the more important function of Na,K-ATPase is the transport of Na+ inside MVs. Now we have included more details about PHOSPHO1 in the introduction. Please see revised version.
It is well established that the Na,K-ATPase enzyme complex consists of two polypeptide chains: the α-subunit or catalytic chain, which is responsible for the enzymatic activity of the complex, and the β-subunit, which has a structural function and may also play a functional role in the catalytic reaction and ion-pumping mechanism. A third γ-subunit is a small hydrophobic proteolipid, which is associated with the Na,K-ATPase, which, although not essential either for catalytic activity or ion transport, acts as a kinetic regulator. The α- and β-subunits also present additional thermal stability that may be modulated by the nature of the co-solvent (detergent or lipid) used in the preparations of the Na,K-ATPase. In addition, distinct processes of β-subunit displacement and α-α-subunit aggregate formation may also contribute to the changes in the CD spectra and the enzyme activity. The authors have not considered how reconstituted enzyme use differs from the detergent-solubilized enzyme.
ANSWSER: Thank you for your comments and observations. We agree with the referee’s observations. Our results in 20 years working with Na,K-ATPases are in synergy with the literature and the referee’s commentaries. For this paper, we chose a successful reconstitution of previously characterized proteoliposomes (Na,K-ATPase-DPPC and Na,K-ATPase-DPPC:DPPE) that considered the functional enzyme in respect to phosphatase activity (not in the context a generating ionic gradient), in a condition which α- and β-subunits are present. Third γ-subunit, is not present, but the protein function is possible in the absence of this subunit, as previously described by various authors and us.
Therefore, to understand the possibility to support mineralization by Pi generation we focused on the solubilized or membrane-reconstituted Na,K-ATPase with phosphatase ability. This result was confirmed in the experiments of the paper with Turbidity (Abs340nm/mg) increase. As other phosphatases, we found that Na,K-ATPase could help to support propagation of mineralization in mimic model but, considering that in the native MVs the Na,K-ATPase is in rightside-out position, lower amounts (ATP hydrolysis site inside of the vesicle) and do not have ATP continuous supply it cannot support or replace the PHOSPHO1 or TNAP. So the putative function of Na,K-ATPase is the transport of Na+ out MVs, that is used by Pit1 Pi cotransporter, in the initial stages of mineralization (sufficient to the nucleation). The Pi produced by Na,K-ATPase adds to the Pi produced by PHSOPSHO1.
Also, the solubilized enzyme is unable to mimic the lipid phase of MVs and assemble/nucleate the phosphates mineral in the absense of the PS-CLPX nucleator, whereas proteoliposomes can.
We politely invite the referee to check our 17 previous publications discussing NKA activity, solubilization, purification and reconstitution that support our work using NKA in this paper.
- Yoneda JS; Sebinelli HG; Itri R; Ciancaglini P. Overview on solubilization and lipid reconstitution of Na,K-ATPase: enzyme kinetic and biophysical characterization. Biophysical Reviews, v. 4, p. 67-81, 2020. doi: 10.1007/s12551-020-00616-5.
- Sebinelli HG; Borin, IA; Ciancaglini P; Bolean M. Topographical and mechanical properties of liposomes surface harboring Na,K-ATPase by means of Atomic Force Microscopy. Soft Matter, v. 15, p. 2737-2745, 2019. doi: 10.1039/c9sm00040b
- Yoneda JS; Scanavachi G; Sebinelli HG; Borges JC; Barbosa LRS; Ciancaglini P; Itri R. Multimeric species in equilibrium in detergent-solubilized Na,K-ATPase. International Journal of Biological Macromolecules, v. 89, p. 238-245, 2016. doi: 10.1016/j.ijbiomac.2016.04.058
- Yoneda JS; Rigos CF; Ciancaglini P. Addition of subunit γ, K+ ions, and lipid restores the thermal stability of solubilized Na,K-ATPase. Archives of Biochemistry And Biophysics, v. 530, p. 93-100, 2013. doi: 10.1016/j.abb.2012.12.022
- Ciancaglini P; Simão AMS; Bolean M; Millán JL; Rigos CF; Yoneda JS; Colhone MC; Stabeli RG. Proteoliposomes in nanobiotechnology. Biophysical Reviews, v. 4, p. 67-81, 2012. doi: 1007/s12551-011-0065-4
- Rigos CF; Santos HL; Yoneda JS; Montich G; Maggio B; Ciancaglini P. Cytoplasmatic domain of Na,K-ATPase ?-subunit is responsible for the aggregation of the enzyme in proteoliposomes. Biophysical Chemistry, v. 146, p. 36-41, 2010. doi: 10.1016/j.bpc.2009.10.002
- Barbosa LRS; Rigos CF; Yoneda JS; Itri R; Ciancaglini P. Unraveling the Na,K-ATPase α 4 Subunit Assembling Induced by Large Amounts of C 12 E 8 by Means of Small-Angle X-ray Scattering. Journal of Physical Chemistry. B, v. 114, p. 11371-11376, 2010. doi: 10.1021/jp1013829
- Rigos CF; Nobre TM; Zaniquelli MED; Ward RJ; Ciancaglini P. Circular Dichroism associated with Surface Tension and Dilatational Elasticity to study the Association of Na,K-ATPase subunits.. Journal of Colloid and Interface Science (Print), v. 325, p. 478-484, 2008. doi: 10.1016/j.jcis.2008.06.011
- Santos HL; Rigos CF; Tedesco AC; Ciancaglini P. Biostimulation of Na,K-ATPase by low-energy laser irradiation (685 nm, 35 mW): comparative effects in membrane, solubilized and DPPC:DPPE-liposome reconstituted enzyme. Journal of Photochemistry and Photobiology. B, Biology, v. 89, p. 22-28, 2007. doi: 10.1016/j.jphotobiol.2007.07.007
- Rigos CF; Santos HL; Ward RJ; Ciancaglini P. Lipid Bilayer Stabilization of the Na,K-ATPase Reconstituted in DPPC/DPPE Liposomes. Cell Biochemistry and Biophysics, USA, v. 44, n.3, p. 438-445, 2006. doi: 10.1385/cbb:44:3:438
- Yoneda JS; Rigos CF; de Lourenço TFA; Sebinelli HG; Ciancaglini P. Na,K-ATPase reconstituted in ternary liposome: The presence of cholesterol affects protein activity and thermal stability. Archives of Biochemistry And Biophysics, v. 546, p. 136-141, 2005. doi: 10.1016/j.colsurfb.2004.12.013
- Santos HL; Rigos CF; Ciancaglini P. Kinetics behaviors of Na,K-ATPase: comparative study of solubilized and DPPC:DPPE-liposome reconstituted. Comparative Biochemistry and Physiology. B, Biochemistry & Molecular Biology, v. 142, n.3-4, p. 309-316, 2006. doi: 10.1016/j.cbpc.2005.11.003
- Santos HL; Lopes M; Maggio B; Ciancaglini P. Na,K-ATPase reconstituted in liposomes: effects of lipid composition on hydrolytic activity and enzyme orientation. Colloids and Surfaces. B, Biointerfaces, v. 41, n.4, p. 239-248, 2005. doi: 10.1016/j.colsurfb.2004.12.013
- Santos HL; Rigos CF; Tedesco AC; Ciancaglini P. Rose Bengal located within liposome do not affect the activity of inside-out oriented Na,K-ATPase. Biochimica et Biophysica Acta. Biomembranes, v. 1715, p. 96-103, 2005. doi: 10.1016/j.bbamem.2005.07.014.
- Santos HL; Ciancaglini P. Kinetic characterization of Na,K-ATPase from rabbit outer renal medulla: properties of the (αβ)2 dimer. Comparative Biochemistry and Physiology B-Biochemistry & Molecular Biology, v. 135, n.3, p. 539-549, 2003. doi: 10.1016/s1096-4959(03)00139-8
- Rigos CF; Santos HL; Thedei G; Ward RJ; Ciancaglini P. Influence of enzyme conformational changes on catalytic activity investigated by circular dichroism spectroscopy. Biochemistry and Molecular Biology Education, USA, v. 31, n.5, p. 329-332, 2003. Doi: 10.1002/bmb.2003.494031050264
- Santos HL; Lamas RP; Ciancaglini P. Solubilization of Na,K-ATPase from outer medulla of rabbit kidney using only C12E8. Brazilian Journal of Medical and Biological Research, v. 35, n.3, p. 277-288, 2002. org/10.1590/S0100-879X2002000300002
Other studies have shown pH dependence of sodium-independent ouabain-sensitive ATP hydrolysis. The optimal range of ouabain dose is 5-10 mM. However, there was a huge difference when used at 2 and 5 mM but not at three mM concentrations. How would the authors explain the discrepancy? Was pH ever considered in the hydrolysis experiments?
ANSWSER: As previously standardized in our research group (Santos et al., 2005 - Colloids and Surfaces. B, Biointerfaces, v. 41, n.4, p. 239-248, 2005. doi: 10.1016/j.colsurfb.2004.12.013). For inhibition studies, the enzyme or proteoliposome were previously incubated in reaction medium at pH 7.5 with ouabain (for 30 minutes) at 4 ◦C, and processed as previously described for determination of residual ATPase activity. These details were added in material and method of the revised version.
Minor issues:
The results section has too much background information.
ANSWSER: We removed some parts in the results section. Please see revised version.
Legends should contain sufficient detail to make the figure easily understood. It is desirable to have expanded legends/footnotes for figures.
ANWSER: thank you for the suggestion. In the revised version we added more details in the figure legends.
Figure 1: Incorrect reference/typo. AFM analysis in Figure 2.
ANSWSER: Sorry for the mistake. We corrected the reference.
The abbreviation cMV needs to be clarified. “The cleavage resulted in cTNAP (hereinafter referred to as TNAP cleaved from MVs) and cMVs (TNAP free MVs).” It would be appropriate to use the acronym cMV for cleaved MVs rather than TNAP-free MVs.
ANSWSER: thank you for your comments. Since in the format of this journal the Materials and Methods appear after the results and discussion we reorganized the descriptions of cMV and cTNAP. Please see revised version.
Conclusions: “Since five mM levamisole, inhibited around 72-69 % of the total ATP hydrolysis,” pl. change this to 69-72%.
ANSWSER: OK, we changed in the sentence.
Suggestion: Consider restructuring the paragraph - Premise first and findings later.
ANSWSER: OK, we restructured some paragraphs. Please see revised version.
Reviewer 2 Report
Mayor comments
1. My major concern is that all the presented data here strongly support the role of TNAP in the mineralization process of MVs. However, the authors claim the role of Na-K-ATPase in the title and in the discussion: "It is tempting to suggest that all ATP inside MVs is hydrolyzed by NKA providing a part of the Pi necessary for the apatite formation, just after their release from hypertrophied chondrocytes. ", which is not supported by any data presented in the article. Inhibition of mineralization by ouabain, the Na-K-ATPase specific inhibitor, could provide such evidence. Though even in the artificial liposomes devoid of TNAP, the authors fail to present the data of ouabain inhibition and just state in the text: " Addition of ouabain did not prevent completely the ATP nucleation process, even by the addition of the detergent, probably due to the occluded ouabain binding site which is in the lumen of proteoliposomes to inhibit the outside active site of NKA. " and " The remaining activity of the NKA-liposomes after the inhibition by Ouabain did not affected the mineral quality, even after the addition of C12E8 to solubilize the vesicles. " Thus, the authors did not present any solid evidence for an ouabain-dependent ATPase activity that could be attributed to NA-K ATPase.
2. Another major issue is that in Figure 3 the inhibitors in combination result to unexpected effects; for instance, Ouabain counteracts the effect of Levamisole, the most potent inhibitor in the MV and cMV system presented here, which possibly reflects an interaction of the two inhibitors. No explanation is provided in the manuscript. Without proper controls, the drug combinations presented in Figure 3 are unsuitable for the analysis.
3. The authors present only very subtle differences between the two investigated artificial liposomes DPPC or DPPC:DPPE containing Na-K-ATPase in the presence and absence of Ouabain, which indicates that in even this artificial system, the effect of Na-K-ATPase is negligible on mineralization. The explanations on the better accessibility of Na-K-ATPase active site are purely speculative, and the authors present no evidence supporting it. Also, no evidence is provided for lines 187-188: "...detergent exposing ATP to all regions of MVs and enabling the measurement of ATP hydrolysis inside MVs" therefore it is speculative and should be toned down, or experimentally proven for example by the liberating of fluorescent compounds entrapped in the vesicles or by increased accessibility of Na-K ATPase in vanadate binding assays.
Minor comments
1. Fig 2 misses panel names
2. wrong reference in line 201 to Table 2
3. SDs missing for certain means in Table 3
4. How is Tf longer than the essay length 120h in Table 3?
5. SBI-425 is favoring mineralization, reflected by increased Umax values in Table 3. This should be explained.
6. Data in Table 4 indicate that solubilized Na-K-ATPase hydrolyzes ATP just as efficiently as in DPPC-DPPE liposomes. This is somewhat counterintuitive for such a large multimembrane-spanning protein that interacts with lipids that extensively modify its stability and ATPase activity https://doi.org/10.1073/pnas.1620799114. An explanation should be provided for this phenomenon.
7. Many sentences are unclear, not precise enough or grammatically incorrect, e.g., line 141-145 ATPase activity, not ATP activity; 216-219; 263-264.
8. Abbreviations are not always resolved at first mention e.g., DLS line 164, PMP in Table 3.
Author Response
Referee #2
- My major concern is that all the presented data here strongly support the role of TNAP in the mineralization process of MVs. However, the authors claim the role of Na-K-ATPase in the title and in the discussion: "It is tempting to suggest that all ATP inside MVs is hydrolyzed by NKA providing a part of the Pi necessary for the apatite formation, just after their release from hypertrophied chondrocytes. ", which is not supported by any data presented in the article. Inhibition of mineralization by ouabain, the Na-K-ATPase specific inhibitor, could provide such evidence. Though even in the artificial liposomes devoid of TNAP, the authors fail to present the data of ouabain inhibition and just state in the text: " Addition of ouabain did not prevent completely the ATP nucleation process, even by the addition of the detergent, probably due to the occluded ouabain binding site which is in the lumen of proteoliposomes to inhibit the outside active site of NKA. " and " The remaining activity of the NKA-liposomes after the inhibition by Ouabain did not affected the mineral quality, even after the addition of C12E8 to solubilize the vesicles. " Thus, the authors did not present any solid evidence for an ouabain-dependent ATPase activity that could be attributed to Na-K ATPase.
ANSWSER: Thank you very for the observation and for bringing this issue. Ee suggested that “all ATP inside MVs is hydrolyzed by NKA providing a part of the Pi necessary for the apatite formation, just after their release from hypertrophied chondrocytes…” is not isolated in this context and the source of Pi is in complementary to the PHOSPHO1 action (see the sequence of the paragraph). Also, for MVs, a 3-7% ouabain inhibition was reported and is supported by literature.
For proteoliposomes, our group has previously described the conditions for NKA inhibition by ouabain, but it was removed from the final version of paper and we are glad for the referee’s alert. We added more information to solve the questions presented by the referee in the text, and some explanation about ouabain inhibitions also here added.
We kindly invite the referee to check our published papers regarding this matter.
Santos HL; Rigos CF; Ciancaglini P. Kinetics behaviors of Na,K-ATPase: comparative study of solubilized and DPPC:DPPE-liposome reconstituted. Comparative Biochemistry and Physiology. B, Biochemistry & Molecular Biology, v. 142, n.3-4, p. 309-316, 2006. doi: 10.1016/j.cbpc.2005.11.003
Santos HL; Lopes M; Maggio B; Ciancaglini P. Na,K-ATPase reconstituted in liposomes: effects of lipid composition on hydrolytic activity and enzyme orientation. Colloids and Surfaces. B, Biointerfaces, v. 41, n.4, p. 239-248, 2005. doi: 10.1016/j.colsurfb.2004.12.013
Santos HL; Ciancaglini P. Kinetic characterization of Na,K-ATPase from rabbit outer renal medulla: properties of the (αβ)2 dimer. Comparative Biochemistry and Physiology B-Biochemistry & Molecular Biology, v. 135, n.3, p. 539-549, 2003. doi: 10.1016/s1096-4959(03)00139-8
- Another major issue is that in Figure 3 the inhibitors in combination result to unexpected effects; for instance, Ouabain counteracts the effect of Levamisole, the most potent inhibitor in the MV and cMV system presented here, which possibly reflects an interaction of the two inhibitors. No explanation is provided in the manuscript. Without proper controls, the drug combinations presented in Figure 3 are unsuitable for the analysis.
ANSWSER: The combined effect of levamisole and Ouabain could be explained in native MVs (Figure 3-A) by the effect by only levamisole in TNAP since the Ouabain not reaches its binding site to inhibit NKA. The increment of the turbidity was observed only in the final hours of the experiment and was poorly significant when compared with levamisole alone. This effect could be explained by unspecific interaction and reducing the effect of levamisole. For example, for cMV (Figure 3-B) this effect was not observed since the contribution of TNAP activity was reduced in 70%. We added this comment in the text.
- The authors present only very subtle differences between the two investigated artificial liposomes DPPC or DPPC:DPPE containing Na-K-ATPase in the presence and absence of Ouabain, which indicates that in even this artificial system, the effect of Na,K-ATPase is negligible on mineralization. The explanations on the better accessibility of Na-K-ATPase active site are purely speculative, and the authors present no evidence supporting it. Also, no evidence is provided for lines 187-188: "...detergent exposing ATP to all regions of MVs and enabling the measurement of ATP hydrolysis inside MVs" therefore it is speculative and should be toned down, or experimentally proven for example by the liberating of fluorescent compounds entrapped in the vesicles or by increased accessibility of Na-K ATPase in vanadate binding assays.
ANSWSER: Yes, we agree with referee that the ATPase activity of Na,K-ATPase is negligible on mineralization. On the other hand since the orientations of the Na,K-ATPase is not related to contribution in the Pi production. Our putative suggestion is in the initial stage of mineralization, to enhanced the concentration of Pi, inside of MVs, by the phosphate transporter 1 (PiT-1) cotrasporting Na+, that in addition to the Pi produced by PHOSPHO1 generates mineral nucleation, forming apatite (HA) See legend of Figure 1. So the function of Na,K-ATPaseis the transport of Na+ inside MVs. We change some parts in the text to clarify this interpretation. Please see revised version.
In additions experiment using fluorescent dye we are used in proteoliposomes in the characterization of the vesicles, please see reference: Santos et al., 2005 - Colloids and Surfaces. B, Biointerfaces, v. 41, n.4, p. 239-248, 2005. doi: 10.1016/j.colsurfb.2004.12.013). At this moment similar experiments with MVs are not possible. Also, from lines 190 to 193 we have shown a consistent decrease in the vesicle’s diameter was achieved by vesicles solubilization by detergent, and therefore the exposure of the enzyme binding sites.
Minor comments
- Fig 2 misses panel names
ANSWSER: Sorry with our mistake, now we add A and B in the Figure2.
- wrong reference in line 201 to Table 2
ANWSER: sorry of this wrong. The reference was changed.
- SDs missing for certain means in Table 3
ANSWSER: OK, we added the ± SD, for the PMP parameter. (See revised version)
- How is Tf longer than the essay length 120h in Table 3?
ANSWSER: Since the curves of Abs340nm/mg versus time were plotted and the sigmoidal tendency of mineral formation was evaluated by the mathematical approach as described by [Genge BR, Wu LNY, Wuthier RE (2007) Kinetic analysis of mineral formation during in vitro modeling of matrix vesicle mineralization: Effect of annexin A5, phosphatidylserine, and type II collagen. Anal Biochem 367:159–166. https://doi.org/10.1016/j.ab.2007.04.029] some curves extrapolates the experimental time realized in the assay.
- SBI-425 is favoring mineralization, reflected by increased Umax values in Table 3. This should be explained.
ANSWSER: The binding site pocket for the uncompetitive inhibitor SBI-425 is most likely not conserved for chicken TNAP, as compared to that in of human or mouse TNAP. Also, ATPase activity assays in Table 1 showed that 10 μM SBI-425 inhibited about 35% of MV total activity whereas 5 mM levamisole inhibited 70%. In this sense, the inhibitory effect cannot be observed in MV, since there is a high abundance of TNAP that cannot be totally inhibited by SBI-425. Whereas, cMV has significant less TNAP (66-92%), therefore the effect of SBI-425 should be minimum and other enzymes replace it activity role (Figure 3). More details were added in the revised version.
- Data in Table 4 indicate that solubilized Na,K-ATPase hydrolyzes ATP just as efficiently as in DPPC-DPPE liposomes. This is somewhat counterintuitive for such a large multimembrane-spanning protein that interacts with lipids that extensively modify its stability and ATPase activity https://doi.org/10.1073/pnas.1620799114. An explanation should be provided for this phenomenon.
ANSWSER: Thank you very much for your comments. Our research group showed the effects of lipidic environment and in the orientation of the reconstituted enzyme. Please see:
Yoneda et al., 2020 doi: 10.1007/s12551-020-00616-5.;
Sebinelli et al., 2019 doi: 10.1039/c9sm00040b;
Yoneda et al., 2013 doi: 10.1016/j.abb.2012.12.022;
Rigos et al., 2006. doi: 10.1385/cbb:44:3:438;
Yoneda et al., 2005 doi: 10.1016/j.colsurfb.2004.12.013;
Santos et al., 2006 doi: 10.1016/j.cbpc.2005.11.003;
Santos et al., 2005 doi: 10.1016/j.colsurfb.2004.12.013).
So, the efficiently hydrolyses by ATP in the DPPC:DPPE-reconstituted enzyme is due the orientation of the enzyme with the ATP hydrolysis oriented out of the vesicle as previously standardized and confirmed by our research group, specifically in Sebinelli et al., 2019 and Santos et al., 2005).
- Many sentences are unclear, not precise enough or grammatically incorrect, e.g., line 141-145 ATPase activity, not ATP activity; 216-219; 263-264.
ANSWSER: Thank you, we rearranged the phrases, please check the revised version
- Abbreviations are not always resolved at first mention e.g., DLS line 164, PMP in Table 3.
ANSWSER: Sorry. Since in the format of this journal the Materials and Methods appear after the results and discussion we reorganized the descriptions and now these abbreviations appears before. Than you, please check the revised version.
Round 2
Reviewer 1 Report
Authors were very responsive to previous critique.
Author Response
Thank you for the opportunity to submit the second revision of our paper “Shedding light on the role of Na,K-ATPase as a phosphatase during matrix vesicle-mediated mineralization” by Heitor Gobbi Sebinelli, Luiz Henrique da Silva Andrilli, Bruno Zocaratto Favarin, Marcos Antonio Eufrasio Cruz, Maytê Bolean, Michele Fiore; Carolina Chieffo; David Magne, Andrea Magrini, Ana Paula Ramos, José Luis Millán, Saida Mebarek, Rene Buchet, Massimo Bottini and Pietro Ciancaglini for a Special Issue entitled “Organic, Inorganic and Natural Molecules in Biomineralization”, in collaboration with the “IJMS”.
The English revision in manuscript was realized, please see revised version with highlights.
We hope that the paper is suitable for publication in the “IJMS” Special Issue entitled “Organic, Inorganic and Natural Molecules in Biomineralization”.
Regards
Professor Pietro Ciancaglini, PhD.
Reviewer 2 Report
The Authors have tackled almost all the raised issues and concerns. The conclusions of the amended manuscript are in better harmony with the presented results. The manuscript is suitable for publishing in its present form.
Author Response

(The authors gave the same response as above.)

Round 3
Reviewer 2 Report
The manuscript has been improved sufficiently and can be published in its present form.